# Negative consequences of failing to communicate uncertainties during a pandemic: an online randomised controlled trial on COVID-19 vaccines

Eleonore Batteux [1], Avri Bilovich,[1] Samuel G B Johnson,[2] David Tuckett[1]

[1]Centre for the Study of Decision-Making Uncertainty, University College London, London, UK
[2]Department of Psychology, University of Warwick, Coventry, UK

**Correspondence to**
Professor David Tuckett;
d.tuckett@ucl.ac.uk

## ABSTRACT

**Objective** To examine the impact of the government communicating uncertainties relating to COVID-19 vaccine effectiveness on vaccination intention and trust after people are exposed to conflicting information.

**Design** Experimental design where participants were randomly allocated to one of two groups.

**Setting** Online.

**Participants** 328 adults from a UK research panel.

**Intervention** Participants received either certain or uncertain communications from a government representative about COVID-19 vaccine effectiveness, before receiving conflicting information about effectiveness.

**Main outcome measures** Vaccination intention and trust in government.

**Results** Compared with those who received the uncertain announcement from the government, participants who received the certain announcement reported a greater loss of vaccination intention (d=0.34, 95% CI (0.12 to 0.56), p=0.002) and trust (d=0.34, 95% CI (0.12 to 0.56), p=0.002) after receiving conflicting information.

**Conclusions** Communicating with certainty about COVID-19 vaccines reduces vaccination intention and trust if conflicting information arises, whereas communicating uncertainties can protect people from the negative impact of exposure to conflicting information. There are likely to be other factors affecting vaccine intentions, which we do not account for in this study.

**Trial registration number** Open Science Framework: https://osf.io/c73px/.

## INTRODUCTION

No decision in healthcare comes without a degree of uncertainty. When recommending a treatment, a medical professional generally knows its effectiveness and possible side effects, along with their associated probabilities, that is, risks. They may also be aware that there is uncertainty surrounding that probability estimate, sometimes called ambiguity or radical uncertainty. This kind of uncertainty is particularly salient in a pandemic, where the precise outcomes of treatments and policies cannot be known. Earlier on in the

### STRENGTHS AND LIMITATIONS OF THIS STUDY

⇒ This study provides experimental evidence of the benefits of communicating with uncertainty rather than certainty during a pandemic.

⇒ Participants were randomly allocated to receive either certain or uncertain hypothetical communications about COVID-19 vaccines.

⇒ Vaccination uptake was measured using a single-item measure of intention.

COVID-19 vaccine roll out, research was still underway to confirm vaccines' effectiveness and risks. Accounts of damaging side effects, such as thrombosis following the AstraZeneca vaccine, severely damaged trust.[1] Today, there remain uncertainties about the effectiveness of vaccines against new variants.

Despite the prevalence of uncertainty, there is a lack of consensus on how best to communicate it.[2] A first step has been to investigate how patients respond to communications of uncertainty, which has largely uncovered negative impacts and led to interrogations on how best to communicate it (if at all).[3] We take a different approach in this paper, where we investigate the negative consequences of failing to communicate uncertainties. Are there times where, however difficult it may be to communicate uncertainties, doing so is better than hiding them? Does failing to communicate uncertainties backfire if people find out they exist and are exposed to conflicting information? We explore these questions by investigating how people respond to conflicting COVID-19 vaccine communications.

### Communicating uncertainty in health

In this paper, we distinguish risk or probabilistic uncertainty (eg, 20% chance of benefit from treatment) from uncertainty, or what can also be referred to as ambiguity. Uncertainty can take various forms: imprecision



(eg, 10–30% chance of benefit from treatment), conflict (eg, experts disagreeing), lack of information (eg, insufficient evidence).[3] All three are present during a pandemic such as COVID-19, so we consider them together here.

Uncertainty is communicated to varying degrees across healthcare. Physicians mention some form of uncertainty in most of their patient encounters, although this tends to be in vague terms (eg, 'there is a chance it will/would not work').[4 5] However, physicians are less likely to report that they would communicate uncertainty if they believe patients will have negative reactions to it, which tends to be the case.[6] Interventions designed to communicate information to patients often include quantitative risk estimates, but mentioning uncertainty tends to be the exception.[2 7] When mentioned, it is usually verbally (eg, 'about' or 'up to'). This highlights the lack of consensus for how and when to communicate uncertainty in health.

This is not surprising given that uncertainty can have negative effects on patients, for both significant (eg, cancer[4]) and more minor (eg, acne[8]) illnesses. Verbal expressions of uncertainty by doctors can lower patient confidence[8] and satisfaction.[4 9] Numerical expressions of uncertainty (eg, ranges) can reduce trust and credibility[10 11] and increase perceptions of risk and worry, although less so when communicated visually compared with textually.[11–13] This could be because people generally think science can provide certainty[14] and therefore interpret expressions of uncertainty as signs of incompetence rather than an inevitable feature of science. Explaining why there is uncertainty might help to mitigate misunderstandings, which has been recommended when communicating uncertainty in general.[15] In addition, providing numerical information about risks and benefits makes patients less likely to overturn their decision in the face of conflicting information.[16]

We focus on the effects of communicating uncertainty in public health, which present notable differences. Discussing uncertainty around numerical risk estimates may not only decrease perceived competence but also increase perceived honesty.[14 17] Although people report preferring to see precision in communications, they would rather uncertainties be disclosed if they exist.[14] This suggests that if people are aware that uncertainties exist, they may be suspicious of communications which do not mention them. Nonetheless, a previous study on vaccine communications during a hypothetical novel pandemic found that uncertain communications led to lower vaccination intention and lower trust in the communicator.[18] However, this may be because the communications were verbal and highly uncertain (eg, 'we are not sure exactly how effective it will be'). There is more precise information in the context of COVID-19, despite prevailing uncertainties.

## What if uncertainties are not communicated?

When uncertainties do exist, can ignoring them backfire? The literature indicates there are advantages to not communicating uncertainties, but it does not address the consequences once people are confronted with information which conflicts with what they were communicated. There are many instances where this applies. A vaccine might be 70% effective against infection, but that does not mean the vaccinated are certain they will not get infected. In contexts where evidence is lacking, new evidence can arise which invalidates previous communications. Although disclosing uncertainties might have negative effects initially, over time it could protect against the consequences of people experiencing undesirable outcomes or conflicting information, which is damaging in science communication.[19]

In other contexts, communicating uncertainty can be beneficial in the long term. In an intelligence context, when people are told a terrorist attack occurred and shown the forecasts, they find forecasters who communicated uncertainty more credible and less worthy of blame.[20] In a geological context, there is no evidence of a difference between certain and uncertain forecasts in terms of perceived correctness and loss of credibility after unlikely events occur.[21] In a financial investment context, when forecasts of future returns turn out to be incorrect, forecasters who did not acknowledge uncertainty were perceived as less trustworthy.[22] Interestingly, this did not lead investors to pull out of their investment, showing that they blame the forecaster for incorrect forecasts but not the object of the forecast. It is worth investigating whether this applies to a medical context, that is, whether failing to communicate uncertainties has worse consequences for confidence in the communicator than in the object of the communication (eg, a treatment or vaccine).

### The present research

We examine how uncertain communications affect trust and vaccination intention over time. Specifically, we test whether communicating uncertainty about COVID-19 vaccines limits any loss of trust and vaccination intention after people receive conflicting information about their effectiveness. We focus on COVID-19 given the need to maximise vaccine uptake, where low trust has been linked to vaccine hesitancy.[23] In addition, COVID-19 provides a real pandemic context that participants can relate to and have knowledge of. Our hypotheses were preregistered on the Open Science Framework[24] and are as follows.

### Hypothesis 1

We expect people are less favourable to vaccination after receiving uncertain compared with certain communications. The first main outcome variable is vaccination intention, which we expect to be lower following uncertain communications, as found in a previous study.[18] We investigate whether this is accompanied by lower perceptions of vaccine effectiveness,[18] stronger avoidance emotions (eg, worry) and weaker approach emotions (eg, excitement). Indeed, emotions are crucial to decision-making in contexts of uncertainty.[25] The second main outcome variable is trust in communicators, which is crucial to both vaccine uptake and compliance to guidelines during

a pandemic.[23 26] Previous studies suggest trust should be lower when uncertainty is communicated.[10 18]

### Hypothesis 2

Once people receive information, which conflicts with earlier communications, we expect those who initially received certain communications to experience more negative effects compared with those who received uncertain communications. We posit that communicating uncertainty makes people more likely to expect information to change over time and therefore less affected when confronted to conflicting information. On the other hand, communicating with unwarranted certainty may be perceived as intentionally misleading. We expect to see greater reductions in vaccination intention in those receiving certain communications.

We conducted a study in November 2020, before COVID-19 effectiveness rates were widely communicated. We presented participants from the general UK population with a hypothetical vaccine announcement containing information about the vaccine's effectiveness, which either conveyed certainty or uncertainty. Participants were then given new information about vaccine effectiveness, which conflicts with the earlier announcement. We compare participants' vaccination intention, trust, perceived vaccine effectiveness and affective reactions after receiving the announcement to after receiving conflicting information.

### METHOD

#### Trial design

Communication certainty (1—certain, 2—uncertain) was manipulated between-subjects.

#### Participants

Overall, 328 participants residing in the UK were recruited using prolific, an online participant recruitment platform (https://www.prolific.co/) (see figure 1). A sample of 328 was required to find a small effect (d=0.20) for hypotheses 2a–e with a mixed model ANOVA (Analysis of Variance) with high power (>0.95) and alpha level (<0.05). This sample size also allows enough power to test hypothesis 1 in accordance with existing findings. Participants were compensated for their time at a rate of £7.50 per hour.

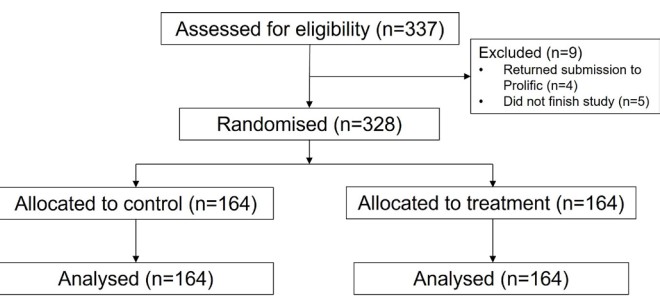

**Figure 1** Consolidated Standards of Reporting Trials flow diagram depicting the phases of participant recruitment and analysis.

They were asked demographic questions (age, gender, level of education). They were then asked questions about COVID-19; first, how much trust they currently have in the government's handling of the COVID-19 crisis on a 5-point scale (1—not at all, 5—a great deal). Second, how reliable, precise and consistent they perceive the science relating to COVID-19 on a 7-point scale (1—reliable/precise/consistent, 7—unreliable/imprecise/inconsistent). These were added to provide an overall score on their perception of the certainty of COVID-19 science. Finally, participants completed the Vaccination Attitudes Examination Scale, which provides an overall score of favorability to vaccination[27] on a 5-point scale (1—strongly disagree, 5—strongly agree). Participant characteristics can be found in table 1.

#### Patient and public involvement

The public was involved in the development of the communications used in the study. We conducted an online pilot study with 50 UK participants to check that the communications about vaccine effectiveness were understandable and believable, with the opportunity for participants to provide feedback.

#### Interventions

Participants were reminded they are in the middle of the COVID-19 pandemic and told to imagine they hear a public health government representative make a vaccine announcement on the news. This announcement states that a vaccine has passed the necessary checks and will soon be available. For those in the certain condition the representative says: "I can confirm that the vaccine is 60% effective. This means that, although the vaccine might not work for everyone, there is a very good chance that it will work for you. This vaccine will significantly drive down the infection rate and we will be able to remove the restrictive measures we put in place to combat the virus." In the uncertain condition the representative says: "The vaccine is between 50% and 70% effective. The reason I can't give a more precise estimate is because the data we have doesn't allow that. There might be some things we don't know yet about the vaccine, but this is the best available option. Although it might not work for everyone, there is a chance it will work for you. This vaccine will hopefully drive down the infection rate and we may be able to remove the restrictive measures we put in place to combat the virus." Then, all participants are told: "a week later, the vaccine is available and you can book an appointment with your local GP practice. Before deciding whether to get it, you want to read the research on the vaccine's effectiveness. You find the latest international piece of research which is deemed to have the most reliable data. This tells you that the vaccine is actually nearer to 40% effective."

#### Outcomes

Measures were taken after participants received the initial announcement and after they read the additional

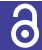

**Table 1** Participant characteristics

| Demographics | |
|---|---|
| Age | M=35.09 (SD=11.36) |
| Gender | 28% male<br>71% female<br>1% non-binary |
| Education | 11% General Certificate of Secondary Education (GCSE) or equivalent<br>23.5% A level or equivalent<br>45% undergraduate degree<br>20% postgraduate degree |
| Beliefs | |
| Trust in government | M=2.13 (SD=0.99) |
| Science certainty | M=11.47 (SD=4.10) |
| Vaccinations | M=39.97 (SD=10.02) |

Trust in government can range from 1 to 5, science certainty from 3 to 21 and vaccination attitudes from 12 to 60 (with higher figures indicating more favourable attitudes to vaccination).

research about the vaccine's effectiveness. Participants were asked how much confidence and trust they have in the government representative, how effective they think the vaccine is, how they feel about getting the vaccine (excited, confident, worried, uncertain) on 5-point scales (1—not at all, 5—a great deal) and how likely they are to get the vaccine on a 5-point scale (1—definitely not, 5—definitely yes).

### Randomisation and blinding

Participants were randomly allocated to the certain or uncertain communication condition via the Qualtrics survey platform randomisation function and were blind to the condition they were allocated to.

### Statistical methods

As specified in the preregistered analysis plan, hypotheses 1 and 2 were tested with mixed model ANOVAs. Announcement certainty (1—certain; 2—uncertain) was a between-subjects factor and time point (1—after

announcement; 2—after conflicting information) was a within-subjects factor. This analysis was conducted for all dependent measures (vaccination intention, effectiveness, trust, confidence, emotions). Outcome assessors were not blind to the treatment group participants were allocated to.

### RESULTS

The findings are broadly consistent across measures of vaccination intention, vaccine effectiveness, trust and confidence in government and emotion. They support the hypothesis that conflicting information leads to more negative effects among those who were exposed to certain compared with uncertain communications (hypothesis 2). However, they do not support the hypothesis that people are initially more favourable to certain compared with uncertain communications (hypothesis 1). Further analyses with demographics and COVID-19-related beliefs are detailed in the online supplemental file 1, which broadly do not affect our findings.

### Vaccination

The certain announcement led to a greater decline in vaccination intention following exposure to conflicting information (see figure 2). There was no difference in vaccination intention between people who received the certain and uncertain announcement after the announcement ($t_{326}$=−0.12, p=0.903, d=0.01, 95% CI (−0.20 to 0.23)), but there was a marginal difference after reading the conflicting information ($t_{326}$=−1.804, p=0.072, d=0.20, 95% CI (0.02 to 0.42)) ($F_{1,326}$=9.50, p=0.002, $\eta_p^2$p20.03). The significant interaction indicates that those who received the certain announcement experienced a greater reduction in vaccination intention than those who received the uncertain announcement. Participants had stronger vaccination intentions after the announcement than after reading conflicting information ($F_{1,326}$=134.47, p<0.001, $\eta_p^2$p20.29) and there was no overall difference between those receiving the certain and uncertain announcement ($F_{1,326}$=1.02, p=0.314, $\eta_p^2$p20.01).

The pattern was the same for effectiveness, where the certain announcement led to a greater decline in perceived effectiveness (see figure 2). After the announcement, perceptions of effectiveness were comparable between those who received the certain and uncertain announcement ($t_{326}$=0.06, p=0.951, d=0.01, 95% CI (−0.23 to 0.21)), whereas those who received the certain announcement perceived it as less effective after reading conflicting information ($t_{326}$=−1.99, p=0.048, d=0.22, 95% CI (−0.00 to 0.44)) ($F_{1,326}$=5.45, p=0.020, $\eta_p^2$p20.02). Participants thought the vaccine was more effective after the announcement than after reading conflicting information ($F_{1,326}$=232.63, p<0.001, $\eta_p^2$p20.42) and there was no overall significant difference between those receiving the certain and uncertain announcement ($F_{1,326}$=1, p=0.318, $\eta_p^2$p20.01).

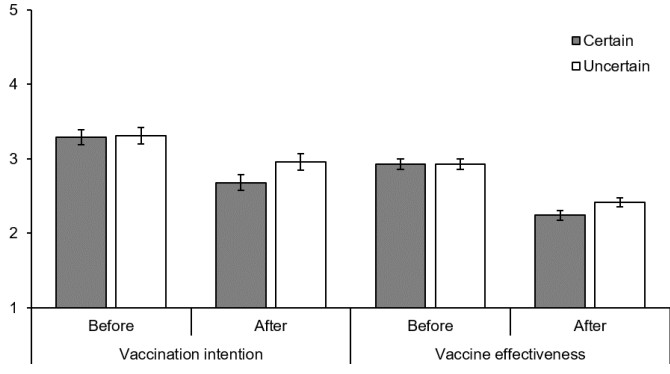

**Figure 2** Vaccination intention and perceived vaccine effectiveness before receiving conflicting information (ie, after the vaccine announcement) and after receiving conflicting information by announcement certainty.

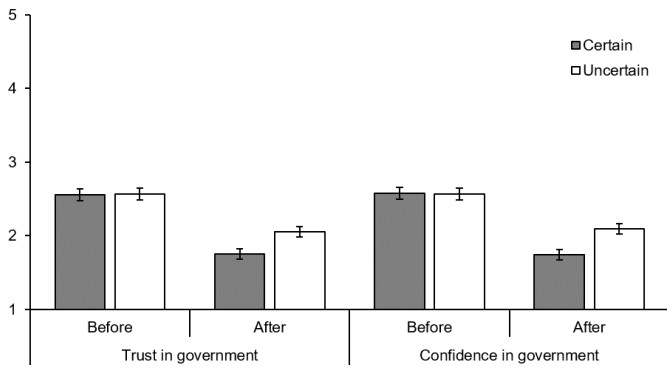

**Figure 3** Trust and confidence in the government representative who made the vaccine announcement before receiving conflicting information (ie, after the vaccine announcement) and after receiving conflicting information by announcement certainty.

### Government

The certain announcement led to a greater decline in trust and confidence in the government representative after exposure to conflicting information (see figure 3). Both groups were equally trusting of the government representative after the announcement ($t_{326}$=−0.54, p=0.957, d=0.01 95% CI (−0.21 to 0.22)), whereas those who received the certain announcement were less trusting after reading conflicting information ($t_{326}$=−3.04, p=0.003, d=0.34, 95% CI (0.12 to 0.55)) ($F_{1,326}$=9.54, p=0.002, $\eta_p^2$p20.03). This interaction means that those who received the certain announcement experienced a greater reduction in trust. Participants had more trust in the government representative after their announcement than after reading conflicting information ($F_{1,326}$=187.12, p<0.001, $\eta_p^2$p20.37) and there was no overall significant difference between those receiving the certain and uncertain announcement ($F_{1,326}$=2.70, p=0.101, $\eta_p^2$p20.01).

This was also the case for confidence (see figure 3). Both groups were equally confident in the government representative after the announcement ($t_{326}$=0.79, p=0.914, d=0.01, 95% CI (−0.23 to 0.21)), whereas those

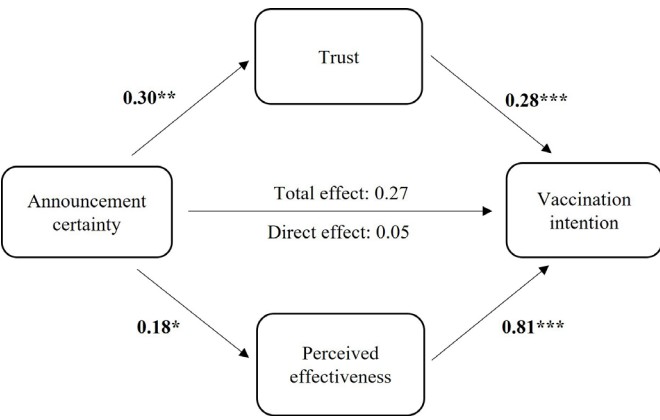

**Figure 4** Relationship between announcement certainty and vaccination intention after receiving conflicting information mediated by trust in government representative and perceived vaccine effectiveness. *p<0.05, **p<0.01, ***p<0.001.

who received the certain announcement were less confident after reading conflicting information ($t_{326}$=−3.45, p=0.001, d=0.38, 95% CI (0.16 to 0.60)) ($F_{1,326}$=12.08, p=0.001, $\eta_p^2$p20.04). This means that those who received the certain announcement experienced a greater reduction in confidence. Participants were more confident in the government representative after their announcement than after reading conflicting information ($F_{1,326}$=170.61, p<0.001, $\eta_p^2$p20.34) and there was no overall significant difference between those receiving the certain and uncertain announcement ($F_{1,326}$=3.13, p=0.078, $\eta_p^2$p20.01).

### Predictors of vaccination intention

In a previous study on communicating uncertainty about vaccines during a pandemic, perceived vaccine effectiveness mediated the relationship between communicated uncertainty and vaccination intention but trust in the government representative did not.[18] We explored whether this was the case here using the PROCESS macro for SPSS[28] (Version 27; see figure 4). Both trust in the government representative (b=0.09, 95% CI (0.02 to 0.18)) and perceived effectiveness (b=0.14, 95% CI (0.003 to 0.29)) mediated the relationship between announcement certainty and vaccination intention. Participants who received the uncertain announcement were more likely to want to get vaccinated, both because they had higher trust in the government representative and because they perceived the vaccine as more effective after receiving conflicting information. Both of these mechanisms contribute to the effect of uncertainty communication on vaccination intention. Trust may not explain the effect of uncertainty communication on vaccination intention when the announcement is made,[18] but it does here after participants are exposed to conflicting information.

### Emotions

Although the pattern of findings on emotions is similar, the differences between those receiving the certain and uncertain announcement were less clear, perhaps due to the hypothetical nature of the study. The certain announcement led to a greater increase in avoidance emotions after exposure to conflicting information (see figure 5). Participants were less worried after the announcement than after reading conflicting information ($F_{1,326}$=60.50, p<0.001, $\eta_p^2$p20.16), which was qualified by an interaction with the certainty of the announcement ($F_{1,326}$=4.86, p=0.028, $\eta_p^2$p20.02). Those who received the certain announcement experienced a greater increase in worry than those who received the uncertain announcement, although there was no statistical difference between each group after receiving the announcement ($t_{326}$=−0.97, p=0.332, d=0.11, 95% CI (−0.11 to 0.32)) or reading the conflicting information ($t_{326}$=0.51, p=0.614, d=0.06, 95% CI (−0.16 to 0.27)). There was no overall significant difference between those receiving the certain and uncertain announcement ($F_{1,326}$=0.05, p=0.819, $\eta_p^2$p20.01).

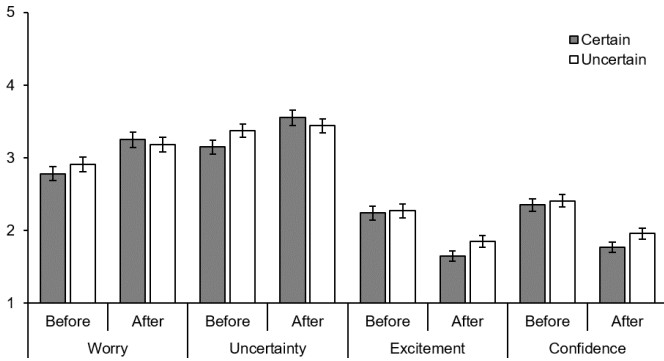

**Figure 5** Emotions before receiving conflicting information (ie, after the vaccine announcement) and after receiving conflicting information by announcement certainty.

Participants were less uncertain after the announcement than after reading conflicting information ($F_{1,326}$=19.35, p<0.001, $\eta_p^2$p20.06), which was qualified by an interaction with the certainty of the announcement ($F_{1,326}$=9.27, p=0.003, $\eta_p^2$p20.03). Those who received the certain announcement experienced a greater increase in uncertainty than those who received the uncertain announcement, although there was no statistical difference between each group after receiving the announcement ($t_{326}$=−1.70, p=0.091, d=0.19, 95% CI (−0.03 to 0.40)) or reading the conflicting information ($t_{326}$=0.74, p=0.462, d=0.08, 95% CI (−0.14 to 0.30)). There was no overall significant difference between those receiving the certain and uncertain announcement ($F_{1,326}$=0.24, p=0.628, $\eta_p^2$p20.01).

The certain announcement did not lead to a greater decrease in approach emotions after conflicting information (see figure 5). Participants were more excited about the vaccine after the announcement than after reading conflicting information ($F_{1,326}$=127.76, p<0.001, $\eta_p^2$p20.28) but the interaction with the certainty of the announcement was marginally significant ($F_{1,326}$=1.20, p=0.060, $\eta_p^2$p20.01). There was no overall significant difference between those receiving the certain and uncertain announcement ($F_{1,326}$=1.05, p=0.306, $\eta_p^2$p20.01). Participants were more confident about the vaccine after the announcement than after reading conflicting information ($F_{1,326}$=126.09, p<0.001, $\eta_p^2$p20.28) but the interaction with the certainty of the announcement was not significant ($F_{1,326}$=2.16, p=0.142, $\eta_p^2$p20.01). There was no overall difference between those receiving the certain and uncertain announcement ($F_{1,326}$=1.41, p=0.235, $\eta_p^2$p20.01).

## DISCUSSION

Communicating uncertainties had protective effects against new conflicting information. Participants showed a greater reduction in vaccination intention after receiving information which conflicted with communications delivered with certainty, as opposed to communications which acknowledged uncertainties. This was accompanied by

a greater reduction in trust in the communicator and perceived vaccine effectiveness, which both affected vaccination intention. Participants also experienced a greater increase in avoidance emotions (worry and uncertainty) following information which conflicted with certain as opposed to uncertain communications. There was no decline in approach emotions, although they were low to begin with.

At the time of the vaccine announcement, we do not find clear evidence that those who received uncertain communications are less likely to get vaccinated. This contrasts with previous findings, although communications in those studies expressed greater uncertainty than here.[18] While most of the previous literature indicates that communicating uncertainty has damaging effects,[3] our findings are an example of the kinds of contexts in which those effects might be weaker, that is, when uncertainty is particularly salient. Patients might not expect scientific uncertainty generally,[14] but people have been exposed to it during COVID-19 and may therefore expect it and want it communicated.[19]

Once people receive information which conflicts with the vaccine announcement, there are differences between those exposed to the certain and uncertain announcement. The government representative who delivered the announcement appears more trustworthy to those who were exposed to uncertainty. Those who received the certain announcement now perceive the vaccine as less effective, although the difference with vaccination intention is less clear. Having said that, those who experience a strong decline in trust and perceived vaccine effectiveness following the certain announcement also experience a strong decline in vaccination intention, making it weaker compared with those who received the uncertain announcement. Although communicating with certainty about vaccines is more damaging for trust in communicators than for vaccination intention, as findings in the financial domain suggested,[22] the effects on vaccination intention remain a problem.

### Limitations

These findings highlight the benefits of communicating uncertainties in health, but they are only a starting point and should be interpreted with caution. This study focused on uncertainties relating to vaccine effectiveness, but there are many other uncertainties relating to vaccines during a novel pandemic worth exploring. Risks of side effects, including those not detectable in rapid trials, are particularly important to the public when making vaccination decisions.[29] Many are motivated to get vaccinated to reduce the spread of the virus and lift restrictions, but whether the vaccination programme can do so is not necessarily known from the outset.[30] We investigated only the influence of government communications on vaccination intention, but there are many other sources of influence, such as medical professionals, friends and family and social media.[31] In addition, we only exposed participants to one instance of conflicting information,

whereas there might be more throughout a pandemic. Vaccination intention and trust are likely to evolve over time and may be more impacted by repeated exposures.

Given the hypothetical nature of the study, caution is warranted when applying findings. We used a hypothetical delay between the vaccine announcement and receiving conflicting information. This makes generalisation to real instances more difficult, given that time delays increase the likelihood that people forget the information they receive and therefore do not interpret new information as conflicting with it. Having said that, government communications and new information are likely to be highly mediatized and conflicts made salient during a crisis like COVID-19.[1] In addition, we used a real pandemic situation where participants had prior knowledge and relevant experiences. They are likely to have been more engaged and invested than in completely hypothetical studies.

It is worth noting that we did not ask participants whether they had been previously diagnosed with COVID-19. It is unclear whether it would have affected their vaccine intentions, although unlikely to be a confound here since participants were randomly assigned to the control and treatment conditions. Previously having had COVID-19 could have made participants feel more strongly about having certainty over vaccine effectiveness due to negative experiences, or less strongly since they could now believe they are immune.

It would be valuable to know how well these findings generalise beyond a pandemic context in the UK. It is worth investigating whether our findings generalise to other situations, such as physician–patient interactions where communicating uncertainty seems initially problematic but may have long-term benefits that have not been uncovered yet. Generalising beyond the UK context would be valuable to inform global communication practices. Given that trust in government is important for vaccine uptake beyond the UK,[23] we expect findings would be similar in other countries.

### Implications for research and policy

These findings highlight the negative consequences of failing to communicate uncertainties. Although communicating with certainty can initially have benefits, if that certainty is not warranted it can have damaging consequences in the long run. Communicators should consider the quality of the evidence and whether people are likely to be exposed to diverging opinions and conflicting information. Anticipating this by discussing uncertainties could avoid negative consequences further down the line. In highly uncertain contexts, people may not actually be averse to uncertainties being communicated, unlike what previous studies in more certain contexts suggest.[3] More work is needed to establish whether people respond differently to uncertain communications depending on the level of contextual uncertainty.

How should uncertainties be communicated? Previous studies suggest some formats are more effective.[12] We used several ways of communicating uncertainty here,

which at present cannot be teased apart. We manipulated the uncertainty of vaccine effectiveness, which was a point estimate in the certain announcement and a range in the uncertain announcement. Ranges may communicate uncertainty but they also increase worry and reduce understanding,[11] suggesting that they alone are not sufficient. We accompanied the range by an explanation for the uncertainty, which could have increased understanding of the uncertainty. We included verbal descriptions of uncertainty regarding the broader risks and benefits of vaccination which may have increased perception of uncertainty, perhaps making participants respond less negatively to conflicting information later on. Future research should evaluate these methods in isolation to better understand their relative effectiveness.

Who is best placed to communicate these uncertainties? This study does not address this, although the following reflections, which could inform future research. People might have different expectations of government compared with medical practitioners given they have particularly low levels of trust in politicians.[32] The effects we find on trust could be due to participants perceiving the government as misleading them into getting vaccinated. People might have different expectations of medical practitioners, including certainty in their communications, thereby reacting negatively to uncertainty. Uncertainty could perhaps be interpreted as incompetence from medical practitioners but honesty from politicians. This suggests there may be instances where governments are better placed to communicate uncertainty, particularly during a national crisis, which further research should clarify. In doing so, it is also worth investigating whether political persuasion and government popularity affects trust in government communications and vaccine intention.

### CONCLUSION

During a novel pandemic, where evidence is lacking and evolves over time, people often face changing and conflicting information. Under these circumstances, we show that the government communicating uncertainties attenuates the negative consequences of being faced with conflicting information. Although it comes with challenges, communicating uncertainty can be beneficial for maintaining trust and patient commitment over time. It takes more account of the potential for healthcare communications to develop active expertise in its recipients, thereby developing shared and resilient understanding.[33 34] Our findings support calls for greater transparency about uncertainty in communications relating to COVID-19.[35 36]

**Acknowledgements** We thank Gerd Gigerenzer for comments on the final manuscript.

**Contributors** The study was conceptualised by EB and AB. EB collected and analysed the data. EB, AB, SGBJ and DT contributed to and approved the final

manuscript. EB acts as a guarantor for the overall content, who accepts full responsibility for the conduct of the study, had access to the data, and controlled the decision to publish.

**Funding** This research has been supported by the Think Forward Initiative (a partnership between ING Bank, Deloitte, Dell Technologies, Amazon Web Services, IBM and the Center for Economic Policy Research). Grant number not applicable. The views and opinions expressed in this paper are solely those of the authors and do not necessarily reflect the official policy or position of the Think Forward Initiative or any of its partners.

**Competing interests** All authors declare: support from the Think Forward Initiative for the funding of the submitted work; no financial relationships with any organisations that might have an interest in the submitted work in the previous three years; no other relationships or activities that could appear to have influenced the submitted work.

**Patient and public involvement** Patients and/or the public were involved in the design, or conduct, or reporting, or dissemination plans of this research. Refer to the Methods section for further details.

**Patient consent for publication** Not applicable.

**Ethics approval** This study involves human participants. Ethical approval for this study was obtained from University College London's Research Ethics Committee (approval number: 15313/001). Participants gave informed consent to participate in the study before taking part.

**Provenance and peer review** Not commissioned; externally peer reviewed.

**Data availability statement** Data are available in a public, open access repository. Anonymised data is available on Open Science Framework (https://osf.io/c73px/).

**ORCID iD**
Eleonore Batteux http://orcid.org/0000-0002-5494-7385

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
