## [Reviewer comments · BMJ Open]

ARTICLE DETAILS

TITLE (PROVISIONAL)	The negative consequences of failing to communicate uncertainties during a pandemic: An online randomized controlled trial on COVID-19 vaccines
AUTHORS	Batteux, Eleonore; Bilovich, Avri; Johnson, Samuel; Tuckett, David

VERSION 1 – REVIEW

REVIEWER	Holly Seale University of New South Wales, School of Public Health and Community Medicine
REVIEW RETURNED	24-May-2021

GENERAL COMMENTS	Introduction 1. I do not feel it is necessary to introduce a gender into the opening paragraph2. In regard to the following; However, physicians are less likely to communicate uncertainty if they...': Is this based on self- report from the physician? If so, it should be framed as such.3. The paragraph outlining the negative effects of uncertainty is useful for setting the scene for the new work undertaken by the authors, however without providing the context for the previous studies, I wonder whether some value is lost. For example, if the previous studies are undertaken focused on uncertainty and its impact on health issues, where the patient is facing an extreme negative outcome, then it would be useful to know. I would assume that the role of uncertainty would be very different in that situation, then in those where the outcome is perhaps not so dire. Methods 4. Did you screen your participants to establish whether they had been previously diagnosed with COVID? Could their recent experiences of COVID have had an influence on their perceptions and responses to the questions?5. What questions about COVID were included in the survey instrument6. Were the participants debriefed after being exposed to the scenario and answering the questions? Results 7. The sentence beginning 'Participants receiving the certain and uncertain...' needs to be revised for clarity.
--

REVIEWER	Calvin Chou UCSF, Medicine
REVIEW RETURNED	05-Oct-2021

GENERAL COMMENTS	I commend the authors for undertaking, in this age of uncertainty in the pandemic, this highly relevant study on how communication,
---

	specifically governmental representatives who may fail to disclose uncertainty, is critical in public health. Several findings are interesting: participants in this study were not favorable to certain communication, especially in light of conflicting information, and that such communication would lead theoretically to a decrease in intention to vaccinate, and decreased trust in governmental officials. Please see below for detailed comments and misgivings. (Page numbers listed for this review correspond to publisher page numbers - of 30 - rather than author pagination.) Introduction P4 line 19: I would modify the statement, as it not only reflects the potential power of uncertain communication that the manuscript describes, but it is also fully true clinically, to: 'when recommending a treatment, a medical professional GENERALLY knows' its effectiveness P5 starting line 27 - perhaps more of a style issue - it seems that this section could be streamlined. The explications here about the background in communicating uncertainty might land better in the discussion section, where the authors can contextualize their findings with the extant literature. The more detailed information here could then be telegraphed more succinctly. P7 starting line 17 - the additional contexts that the authors raise are of interest and relevant. Methods - P9 line 31: interesting means of determining uncertainty numerically (adding trust with perception of science). It makes sense as I think of it. Is there validation for this approach? P10 line 19 (Scenario): there appears to be a bit of a leap in the conditions of the study here: the authors study if the participants were solely to rely on a government representative's reports, whether there would be a change in vaccination intention. But as we know, there are numerous inputs to whether someone intends to vaccinate. The authors address this in part later in the Discussion, where medical practitioners may have higher trust than government; there are also many other influences, including, in this day and age, social media. Table 1: does the preponderance of female respondents change the conclusion? Perhaps men are more favorable to certainty in communication even if it is wrong? One other potentially confounding demographic that seems germane, not addressed by the authors, is political persuasion. Results Unfortunately, I cannot read Figure 3 on my pdf. Discussion Limitations section nicely highlights the importance of potential side effects in the ultimate decision to undergo vaccination, and the hypothetical delay. Please see above for other possible further limitations.
--	--

REVIEWER	Thomas Mathew University of Maryland Baltimore County
REVIEW RETURNED	17-Dec-2021

GENERAL COMMENTS	This article reports the design of an online experiment, and the results from the data analysis, in order to assess the negative
--

	consequences resulting from the failure to properly communicate uncertainties during a pandemic. The focus of the experiment is COVID-19 vaccines, and the outcome measures of interest are vaccination intention and trust in government. Being a statistician, I evaluated the article from a statistical perspective only. I found the statistical methodology part quite sound. I also noticed that most of the conclusions in the Results section are quite clear-cut, since the p-values are mostly quite small or rather large. In other words, they were not marginally significant, or marginally insignificant. My only comment here is that I would have liked to see a bit more detail on the mixed model ANOVA that the authors have used, mentioned on p. 8 in the Method section. There is only a mention of this in this section. Someone who wants to reproduce the results, or carry out a similar analysis, would like to know what are the effects in the model, and if the model fit is adequate. I assume the random effect in the model correspond to the participants. It will be helpful if the authors can briefly mention such details, perhaps in the Appendix. In the manuscript that I accessed for review, Figure 3 is not visible. It is just a dark box above the figure caption.
--	---

VERSION 1 – AUTHOR RESPONSE

Reviewer: 1

Dr. Holly Seale, University of New South Wales

Comments to the Author:

Introduction

1. I do not feel it is necessary to introduce a gender into the opening paragraph.

This has been changed to 'they'.

2. In regard to the following; However, physicians are less likely to communicate uncertainty if they...': Is this based on self- report from the physician? If so, it should be framed as such.

This is based on self-reports and has now been worded as such on page 3.

3. The paragraph outlining the negative effects of uncertainty is useful for setting the scene for the new work undertaken by the authors, however without providing the context for the previous studies, I wonder whether some value is lost. For example, if the previous studies are undertaken focused on uncertainty and its impact on health issues, where the patient is facing an extreme negative outcome, then it would be useful to know. I would assume that the role of uncertainty would be very different in that situation, then in those where the outcome is perhaps not so dire.

Thank you for highlighting this. The papers we have cited have found negative effects for a range of contexts/illnesses, such as cancer, GP interactions and acne medication. We have specified this now on page 4: "This is not surprising given that uncertainty can have negative effects on patients, for both significant (e.g. cancer) and more minor (e.g. acne) illnesses."

Methods

4. Did you screen your participants to establish whether they had been previously diagnosed with COVID? Could their recent experiences of COVID have had an influence on their perceptions and responses to the questions?

We did not screen participants to establish whether they had been previously diagnosed with COVID-19. It is unclear whether a recent COVID-19 experience would have affected the results. It could have made participants feel more strongly about having an effective vaccine, or less strongly since they could now believe they are immune. It is not clear whether either would have affected perceptions of

uncertainty. Either way, since participants were randomly assigned to the control (certain announcement) and treatment groups (uncertain announcement), any effect of COVID diagnosis or experience would not confound the results.

5. What questions about COVID were included in the survey instrument.

The questions about COVID-19 are included on page 7, although we have revised the wording to make clear which questions we are referring to. These were questions about trust in the government's handling of COVID-19 and perceptions of the science relating to COVID-19.

6. Were the participants debriefed after being exposed to the scenario and answering the questions?

Participants were debriefed at the end of the study and told that the study was investigating how people respond to different ways of presenting information about COVID-19 vaccines and tested whether the certainty and confidence of predictions reduce confidence in health communications if they turn out to be wrong. Participants were given an opportunity to provide feedback about the study.

Results

7. The sentence beginning 'Participants receiving the certain and uncertain...' needs to be revised for clarity.

Thank you for spotting this. It has now been clarified: "After the announcement, perceptions of effectiveness were comparable between those who received the certain and uncertain announcement ($t_{326}=0.06$, $p=.951$, $d=0.01$ 95% CI [-0.23, 0.21]), whereas those who received the certain announcement perceived it as less effective after reading conflicting information ($t_{326}=-1.99$, $p=.048$, $d=0.22$ 95% CI [-0.00, 0.44]) ($F_{1,326}=5.45$, $p=.020$, $\eta^2=0.02$)."

Reviewer: 2

Dr. Calvin Chou, UCSF

Comments to the Author:

I commend the authors for undertaking, in this age of uncertainty in the pandemic, this highly relevant study on how communication, specifically governmental representatives who may fail to disclose uncertainty, is critical in public health. Several findings are interesting: participants in this study were not favorable to certain communication, especially in light of conflicting information, and that such communication would lead theoretically to a decrease in intention to vaccinate, and decreased trust in governmental officials.

Please see below for detailed comments and misgivings.

(Page numbers listed for this review correspond to publisher page numbers - of 30 - rather than author pagination.)

Introduction

P4 line 19: I would modify the statement, as it not only reflects the potential power of uncertain communication that the manuscript describes, but it is also fully true clinically, to: 'when recommending a treatment, a medical professional GENERALLY knows' its effectiveness

Thank you for the suggestion. We have amended this.

P5 starting line 27 - perhaps more of a style issue - it seems that this section could be streamlined. The explications here about the background in communicating uncertainty might land better in the discussion section, where the authors can contextualize their findings with the extant literature. The more detailed information here could then be telegraphed more succinctly.

We have shortened this section, as well as the introduction generally, to make it more streamlined and targeted to our study.

P7 starting line 17 - the additional contexts that the authors raise are of interest and relevant. We have kept this section when shortening the introduction.

Methods -

P9 line 31: interesting means of determining uncertainty numerically (adding trust with perception of science). It makes sense as I think of it. Is there validation for this approach?

As we outline in our introduction (top of page 4), communicating uncertainty numerically is the exception rather than the norm in a medical context. However, uncertainty is communicated numerically more often in other contexts, such as finance (Du, Budescu, Shelly & Omer, 2011, OBHDP) and climate science (Spiegelhalter, Pearson & Short, 2011, Science).

P10 line 19 (Scenario): there appears to be a bit of a leap in the conditions of the study here: the authors study if the participants were solely to rely on a government representative's reports, whether there would be a change in vaccination intention. But as we know, there are numerous inputs to whether someone intends to vaccinate. The authors address this in part later in the Discussion, where medical practitioners may have higher trust than government; there are also many other influences, including, in this day and age, social media.

Thank you for highlighting this. We have now mentioned this in the limitations section of the discussion on page 15: "We investigated only the influence of government communications on vaccination intention, but there are many other sources of influence, such as medical professionals, friends and family and social media."

Table 1: does the preponderance of female respondents change the conclusion? Perhaps men are more favorable to certainty in communication even if it is wrong?

Thank you for highlighting this. We have checked whether changes in vaccination intention in response to certain vs uncertain communications were affected by gender, but do not find that this is the case. There was no main effect of gender ($F=3.56$, $p=.060$, $\eta^2=0.01$), no interaction between gender and time of announcement ($F=0.59$, $p=.442$, $\eta^2<0.01$), no interaction between gender and certainty of announcement ($F=0.93$, $p=.760$, $\eta^2<0.01$), and no interaction between gender, time of announcement and certainty of announcement ($F<0.01$, $p=.990$, $\eta^2<0.01$). In addition, in the Supplementary File, we included an analysis of the differences in outcome before and after conflicting information. Although we find that gender has an effect for effectiveness and trust, the effects of announcement certainty are still very clear, suggesting that any effect of gender does not change the conclusion.

One other potentially confounding demographic that seems germane, not addressed by the authors, is political persuasion.

We have mentioned this when discussing whether governments are better suited to communicating uncertainty during a public health crisis: "It is also worth investigating whether political persuasion and government popularity affects trust in government communications and vaccine intention."

Results

Unfortunately, I cannot read Figure 3 on my pdf.

This has been fixed.

Discussion

Limitations section nicely highlights the importance of potential side effects in the ultimate decision to undergo vaccination, and the hypothetical delay. Please see above for other possible further limitations.

Thank you for the suggestions above, which have been added to the discussion.

Reviewer: 3

Dr. Thomas Mathew, University of Maryland Baltimore County

Comments to the Author:

This article reports the design of an online experiment, and the results from the data analysis, in order to assess the negative consequences resulting from the failure to properly communicate uncertainties during a pandemic. The focus of the experiment is COVID-19 vaccines, and the outcome measures of interest are vaccination intention and trust in government.

Being a statistician, I evaluated the article from a statistical perspective only. I found the statistical methodology part quite sound. I also noticed that most of the conclusions in the Results section are quite clear-cut, since the p-values are mostly quite small or rather large. In other words, they were not marginally significant, or marginally insignificant. My only comment here is that I would have liked to see a bit more detail on the mixed model ANOVA that the authors have used, mentioned on p. 8 in the Method section. There is only a mention of this in this section. Someone who wants to reproduce the results, or carry out a similar analysis, would like to know what are the effects in the model, and if the model fit is adequate. I assume the random effect in the model correspond to the participants. It will be helpful if the authors can briefly mention such details, perhaps in the Appendix.

Thank you for carefully evaluating our statistics and for the suggestion. This has now been detailed on page 8 with an analysis section in the method section.

In the manuscript that I accessed for review, Figure 3 is not visible. It is just a dark box above the figure caption.

This has been fixed.

VERSION 2 – REVIEW

REVIEWER	Thomas Mathew University of Maryland Baltimore County
REVIEW RETURNED	19-Apr-2022
GENERAL COMMENTS	No further comments.